**Identifying a Transition Climate Zone in an Arid River Basin using Evaporative Stress**
**Index**
Yongqiang Liu[1], Lu Hao[2], Decheng Zhou[2], Cen Pan[2], Peilong Liu[2], Zhe
Xiong[3], Ge Sun[4]
[1]Center for Forest Disturbance Science, USDA Forest Service, Athens, Georgia, USA
[2]Jiangsu Key Laboratory of Agricultural Meteorology, International Center for  Meteorology,
Ecology, and Environment, College of Applied Meteorology, Nanjing University of
Information Science and Technology, Nanjing, China
[3]Institute of Atmospheric Physics, Chinese Academy of Sciences, Beijing, China
[4]Eastern Forest Environmental Threat Assessment Center, USDA Forest Service, Raleigh,
North Carolina, USA
*Correspondence to*: Yongqiang Liu (yongqiang.liu@usda.gov), Lu Hao
**Abstract.** Aridity indices have been widely used in climate classification. However, there is
not enough evidence for their ability in identifying the multiple climate types in areas with
complex topography and landscape, especially in those areas with a transition climate. This
study compares a traditional meteorological aridity index (*AI*), defined as the ratio of precipitation
(*P*) to potential evapotranspiration (*PET*), with a hydrological aridity index, the
Evaporative Stress Index (*ESI*) defined as the ratio of actual evapotranspiration (*AET*) to *PET*.
in the Heihe River Basin (HRB) of the arid northwestern China. *PET* was estimated using the
Penman-Monteith and Hamon methods. The aridity indices were calculated using the high
resolution climate data simulated with a regional climate model for the period of 1980-2010.
The climate classified by *AI* shows a climate type for the upper basin and a second type for the middle
and lower basin, while three different climate types are found using ESI, each for  one
river basin, indicating that only *ESI* is able to identify a transition climate zone in the middle
basin. The difference between the two indices is also seen in the inter-annual variability and
extreme dry / wet events. The magnitude of variability in the middle basin is close to that in
the lower basin for *AI*, but different for *ESI*. *AI* had larger magnitude of the relative inter-annual
variability and greater decreasing rate from 1980-2010 than *ESI*, suggesting the role of local
hydrological processes in moderating extreme climate events. Thus, the hydrological aridity
index is better than the meteorological aridity index for climate classification in the arid Heihe
River Basin.

## 1 Introduction

The Koppen climate classification is the most widely used climate classification system at large geographic scales. The climate classification is constructed based on the properties of ecosystems, latitude, and average and seasonal precipitation and temperature (Peel et al., 2007). Aridity indices, which combine one or several variables (indicators) into a single numerical value to measure water deficit over long periods (e.g., 30 years or longer) (Wilhite and Glantz, 1985, Zargar et al., 2011), are another useful tool for climate classification (https://en.wikipedia.org/wiki/Climate_classificationin).

Aridity indices can be categorized into different types including meteorological and hydrological indices, which could be simply considered as a lack of water due to anomalous atmospheric and land-surface conditions, respectively. Precipitation, temperature and humidity are atmospheric conditions often used to estimate meteorological aridity indices. The earliest aridity index, developed more than a century ago, reflects the effects of the thermal regime and the amount and distribution of precipitation in determining the native vegetation possible in an area. By the middle of the $20^{th}$ century, attentions turned to precipitation and potential evaporation (Huschke, 1959). The Budyko-type aridity index (AI) (Budyko, 1974), for example, uses annual averages of precipitation and potential evapotranspiration (PET), which is mainly determined by temperature.

Land-surface conditions such as streamflow, runoff, actual evapotranspiration, etc. are variables often used in hydrological aridity indices (Maliva and Missimer, 2012). The Evaporative Stress Index (ESI), for example, defines dryness degree based on the ratio of actual evapotranspiration (AET) to PET over both short and long periods. A relatively low ESI indicates water limitation to plants and the actual rate is way below the PET. In contrast, a relatively high ESI indicates freely available water with the AET rate approaching or close to the PET. The ESI has been long used to evaluate the irrigation need for crop growth and land classification (Yao, 1974). The ESI has been used recently to evaluate water stress using remotely sensed hydrological and ecological properties (Anderson et al., 2016).

There are many similarities between aridity indices and drought indices that measure water deficit over short periods (such as months, seasons, and years). Drought indices also are categorized into meteorological, hydrological, and other types of indices. Percent of Normal (PN) and Standardized Precipitation Index (SPI) (McKee et al., 1993) are simply based on precipitation and can be used to

measure anomalies of a period over various lengths. Palmer Drought Severity Index (PDSI) (Palmer, 1965) and Keetch-Byram Index model (Keetch and Byram, 1968) are based on water supply and demand estimated mainly using precipitation and temperature (Guttman, 1999). Both PDSI and KBDI depend on precedent daily or monthly values, making them specifically useful for a persistent event like drought. Among various hydrological drought indices, Streamflow Drought Index (SDI) (Nalbantis and Tsakiris, 2009) and Surface Water Supply Index (SWSI) (Shafer and Dezma, 1982) use streamflow as well as reservoir storage and precipitation to monitor abnormal surface water (Narasimhan and Srinivasan, 2005). Standardize Runoff Index (SRI) (Shukla and Wood, 2008) is standard normal deviate associated with runoff accumulated over a specific duration.

Large river basins at continental and sub-continental scales usually encompass multiple climate types related to complex topography and landscape. Climate is more humid in the upper basin near the river origins with high elevations and forest and / or permanent snow cover than the lower basin with low elevations and less vegetated lands. Climate could be extremely dry in parts of a watershed under a prevailing atmospheric high-pressure system. The sub-continental Colorado River watershed, for example, is dominated by cold and humid continental climate in the upper basin of the Rocky Mountains and cold semi-arid or warm desert climate in the lower basin of the southern inter-mountains.

This feature of multiple climate types is also seen in some smaller basins. The Heihe River Basin (HRB) in northwestern China, for example, has an area of 130, 000 $km^2$ with annual precipitation varying dramatically from about 500 mm in the upper basin of the Qilian Mountains with forest-meadow-ice covers in the south to less than 100 mm in the lower basin of the Alxa High Plain with Gobi and sandy lands in the north. Climate types change from cold and humid continental to arid desert, accordingly.

The relative high precipitation in the humid upper basin supports forests and meadows and provides source water lower reach of the Heihe River. In contrast, water is a major limitation factor in arid lower basin. In addition, more extreme weather conditions, especially droughts, occur in arid lower basin. In the Colorado River basins, the reconstructed data show decadal periods of persistently low flows during the past centuries (Woodhouse et al., 2010). The

drought severity in the new millennia has been the most extreme over a century (Cayan et al.,
2010). The reconstructed precipitation series in the HRB indicates that droughts were much
more frequent and lasted longer than floods in the past two centuries (Ren et al., 2010).
Droughts occurred more often in the dry lower basin than the humid upper basin (Li, 2012).

The watersheds with varied topography and landscape may have a transition climate zone
between the two zones. In the HRB, for example, the Koppen climate classification, one of the
most widely used climate classification techniques at large geographic scales and constructed
based on the properties of ecosystems, latitude, and average and seasonal precipitation and
temperature, shows polar tundra or boreal climate in the upper basin of the mountain regions
in the south, arid desert climate in the lower basin in the north, and a transition zone of steppe
climate in the middle. Identifying this transition zone and understanding its unique climate
features are of both scientific and management significance. The complex topography in upper
basin and harsh climate in lower basin make both regions unsuitable for human living. The
transition zone however is relatively flat in comparison with the mountain region and less arid
in comparison with the dryland region. It therefore provides a favorable condition for industrial
and agricultural development. Also, the environmental conditions in this region are more
dynamical and localized because of human induced rapid and fragmental landscape changes.

ESI is a newly developed aridity, which is similar to AI but more related to surface hydrology.
However, ESI applications for climate classification have yet been conducted. In addition, many
studies have compared ESI with other drought indices in different climatic environments. Otkin et al.
(2013) compared the ESI with drought classification used by the U.S. Drought Monitor (USDM)
(Svoboda et al., 2002) and found that the ESI anomalies led the USDM drought depiction by several
weeks and large ESI anomalies therefore were indicative of rapidly drying conditions. This finding was
coincident with the droughts occurred across the United States in recent years. Choi et al. (2013)
compared the ESI with the Palmer drought severity index (PDSI) in a watershed of the Savannah River
branch in southeastern United States during 2000-2008. They found that the ability of the ESI to
capture shorter term droughts was equal or superior to the PDSI when characterizing droughts for the
watershed with a relatively flat topography dominated by a single land cover type. However, the
differences between the meteorological and hydrological indices in capturing the spatial
patterns under complex topography and environments, especially with a transition zone,
are not well characterized and understood.

This study is to understand the capacity of the meteorological aridity index, AI, and the hydrological
aridity index, ESI, in climate classification, especially in identifying the transition climate zone in
the HRB. The analysis of the transition climate zone was made by comparing the spatial patterns and
regional averages. Their temporal variations were also analyzed to understand the differences in the
seasonal and inter-annual variability and long-term between ESI and AI. These two indices
reflect the water (precipitation and evapotranspiration) and heat (radiation) properties on
the ground surface without the needs to obtain the complex vegetation and soil
hydrological properties. The surface properties needed to calculate ESI and AI could be
obtained from regional climate modeling, which was an approach used in this study.

**2 Methods**
**2.1 Study region**

The study region was the HRB and the adjacent areas (Fig. 1). The Heihe River origins from
the Qilian Mountains in the northern edge of the Tibet Plateau and flows northward to the
China-Russian border. The HRB spans between 98°~101°30′E and 38°~42°N. The upper HRB
is within the mountains elevated 2300~3200m mainly covered with forests and mountain
meadows. The middle HRB is along the Hexi Corridor elevated 1600~2300m mainly covered
with piedmont steppe grass, crops, and residence and commercial uses. The lower HRB is in
the Alxa High-Plain elevated below 1600m mainly covered with Gobi and desert sands.

Annual precipitation is over 400mm in the upper basin, with the maximum of 800mm at
extremely high elevations, about 100~250mm in the middle basin, and below 50mm in many
lower basin areas. The annual precipitation in the upper basin has high seasonal variability,
and nearly 70% of the total annual rainfall occurs from May to September (Gao et al., 2016).
The upper basin generates nearly 70% of the total river runoff, which supplies agricultural
irrigation and benefits the social economy development in the middle and lower basin reaches
(Yang et al., 2015; Chen et al., 2005). Annual mean temperature is about $-4^{\rm O}$C in the upper basin,
$7^{\rm O}$C in the middle basin, and nearly $9^{\rm O}$C in the lower basin.

**2.2 Aridity indices**

The meteorological aridity index is defined as $AI = P / PET$, where $P$ and $PET$ are daily
precipitation and potential evapotranspiration, respectively. $AI$ is a variant of the index
originally defined by Budyko (1974), which is the ratio of annual $PET$ to $P$. The average $AI$
values were used to classify the arid, semi-arid, semi-humid (sub-humid), and humid climate
with the ranges of $AI \leq 0.2$, $0.2 < AI \leq 0.5$, $0.5 < AI \leq 1.3$, and $AI > 1.3$, respectively (Ponce et al.,

169  2000).


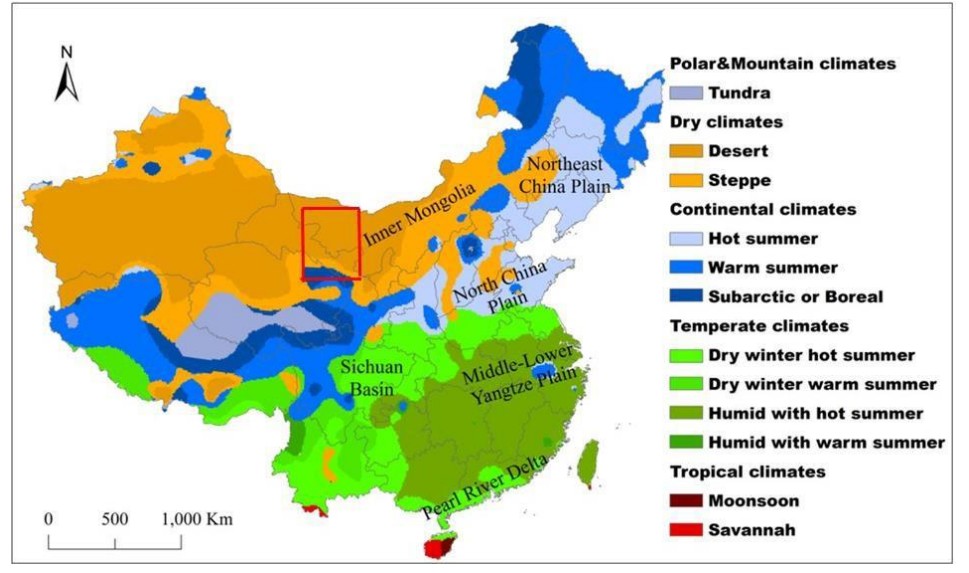

Figure 1. The study region of the Heihe River Basin (red box) in China and the Koppen climate
classification (from Peel et al., 2007).

The hydrological aridity index is defined as $ESI = AET / PET$, where $AET$ is daily actual
evapotranspiration. The ranges of average ESI values of $ESI \leq 0.1$, $0.1 < ESI \leq 0.3$, $0.3 < ESI$
$\leq 0.6$, and $ESI > 0.6$ were used to classify the arid, semi-arid, semi-humid, and humid climate,
respectively (Yang, 2007). This approach agrees with Anderson (2011), which showed that the
$ESI$ values varying gradually from 0 to 1 correspond to several USDM drought levels from
exceptional to no drought for each month from April to September across the continental U.S.

Two methods were used to estimate *PET* (mm/d). One was the energy balance based FAO-
56 Penman-Monteith Equation (Allen et al. 1998):

$$PET_p = \frac{0.408\Delta(R_n-G)+\gamma\frac{900}{T+273}u_2(e_s-e)}{\Delta+\gamma(1+0.34u_2)}$$  (1)

where $R_n$ and $G$ are net radiation and soil flux on the ground (MJm$^{-2}$d$^{-1}$); $T$ is air temperature
(°C); $e_s$ and $e$ are saturation and actual water vapor pressure (kPa); $u_2$ is wind speed at 2m above
the ground (ms$^{-1}$); $\Delta$ is the rate of change of $e_s$ with respect to $T$ (kPa/°C); $\gamma$ is the psychrometric
constant (kPa/°C). The other method is the temperature based on Hamon formula (Hamon,

215    1963):


217         $PET_h = (k\times0.165\times216.7\times N \times e_s) / (T+273.3)$  (2)


where $k$ is proportionality coefficient = 1; $N$ is daytime length. $e_s$ is in 100 Pa here.

Monthly *PET*, precipitation and actual evapotranspiration, obtained based on daily values,
were used to calculate the aridity indices. It was assumed that daily *PET*=0 if daily T<0°C.
Their monthly *PET* was not used if *PET*=0 for more than 10 days in a month. In this case, no
aridity indices were calculated for the month. It was also assumed that daily ground energy
was in balance, so $R_n-G=H+L\times AET$, where $H$ and $L$ are sensible heat flux and potential heat
constant.

T-test was conducted to obtain statistical significance of the differences in the aridity index values
between two Heihe River reaches. The data used in calculation and evaluation of the aridity indices
are listed in Table 1.
Table 1. The data used in calculation and evaluation of the aridity indices. H, AET, P, T, and
e (RH) are sensible heat flux, actual evapotranspiration, precipitation, temperature, wind speed,
and water vapor pressure (relative humidity). HRB stands for Heihe River Basin.

| Source | Parameter | Time Period | Space | Reference |
|---|---|---|---|---|
| Simulation | H, AET, P,T, u, e | 1980-2010, daily | HRB, 3 km resolution | Xiong and Yan (2013) |
| Observation | P,T,RH | 1980-2010, daily | 3 sites in HRB | China National Met Sci Infrastructure (data.cma.cn) |


**2.3 Regional climate modeling**


The climatic and hydrological data used to calculate the aridity indices were created from a
regional climate modeling using the Regional Integrated Environmental Model System
(RIEMS 2.0) (Xiong and Yan, 2013). The simulation was conducted over the period of 1980-
2010. The horizontal spatial resolution was 3km. A unique feature with this simulation was
that the model's parameters, including soil hydrological properties, were recalibrated based on
observations and remote sensing data over the HRB that greatly improved the model's
performance. The model evaluation indicated that the model was able to reproduce the spatial
pattern and seasonal cycle of precipitation and surface $T$. The correlation coefficients between
the simulated and observed pentad $P$ were 0.81, 0.51, and 0.7 in the upper, middle, and lower
HRB regions, respectively ($p<0.01$).

The historical $T$ and $P$ observations during the simulation period at Yeilangou of the upper
basin (38.25$^{o}$N, 99.35$^{o}$E, 3300 m above the sea level), Zhangye of the middle basin (38.11$^{o}$N,
100.15$^{o}$E, 1484m), and Dingqing of the lower basin (40.3$^{o}$N, 99.52$^{o}$E, 1177m) were used to
compare with the simulations. We also calculated SPI based on observed precipitation using a built-in
function of the NCAR NCL (https://www.ncl.ucar.edu/). The results with measured precipitation
were used to evaluate the model performance in simulating drought conditions.

**3 Results**


**3.1 Simulated climate and hydrology**


The spatial pattern of the simulated annual $T$ averaged over the simulation period is featured
by the large changes between basin reaches, increasing from about -15$^O$C in the tall mountains
of the upper basin to over 10$^O$C in the deserts of the lower basin (Fig. 2). The simulated average
annual $P$ shows an opposite gradient, decreasing from about 2.5 mm/d in the mountains to less
than 0.25 mm/d in the deserts (Fig. 2). The simulated net radiation decreases from west to east in the
mountains where there is an increasing trend in precipitation. The net radiation is small in the northeastern
section of the domain, probably due to large outgoing long-wave radiation related to clear and relative hot
weather. The simulated average annual $AET$ has a similar pattern to precipitation (Fig. 2). The spatial
variability is much larger within the upper basin than the lower basin.

An interesting feature is that both $T$ and $P$ in the middle basin are very close to their
corresponding values in the lower basin but much different from those in the upper basin; the
$AET$ difference between the middle and upper basin reaches however is much small.

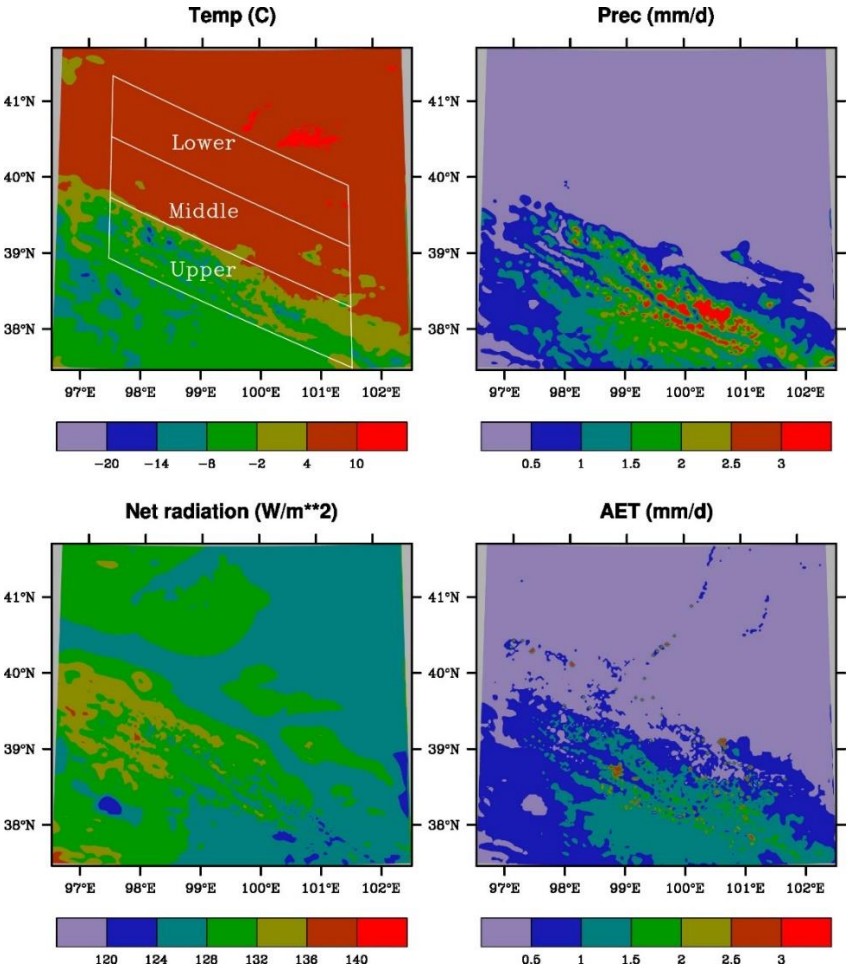



Figure 2. Spatial distributions of simulated air temperature ($T$, $^{o}$C), precipitation (P, mm/d),
net radiation (NRAD, W/m$^2$) and actual evapotranspiration (*AET,* mm/d) averaged over 1980-2010.
The Heihe River basins are shown in the left panel.

As expected, the regional *AET* values averaged over the simulation period are higher in
summer than in winter (Fig. 3). In the upper basin, for example, $T$ increases from about -15$^{o}$C
in winter to 10$^{o}$C in summer, $P$ increased from about 0.25 to 4 mm/d, and *AET* from about 0.25
to 2.5 mm/d. Again, $T$ and $P$ are close between the middle and lower basin reaches all seasons,
and *AET* is close between the middle and upper basin reaches during winter and spring. While
*AET* is close between the middle and lower basin reaches during summer and fall, the differences
between the middle and upper basin reaches are much smaller than the differences in $T$ or $P$. Net
radiation has s seasonal cycle similar to that of temperature. The changing trends among the three
basin reaches are the same between T and NRAD in Spring and Summer but opposite in Winter and
Fall.

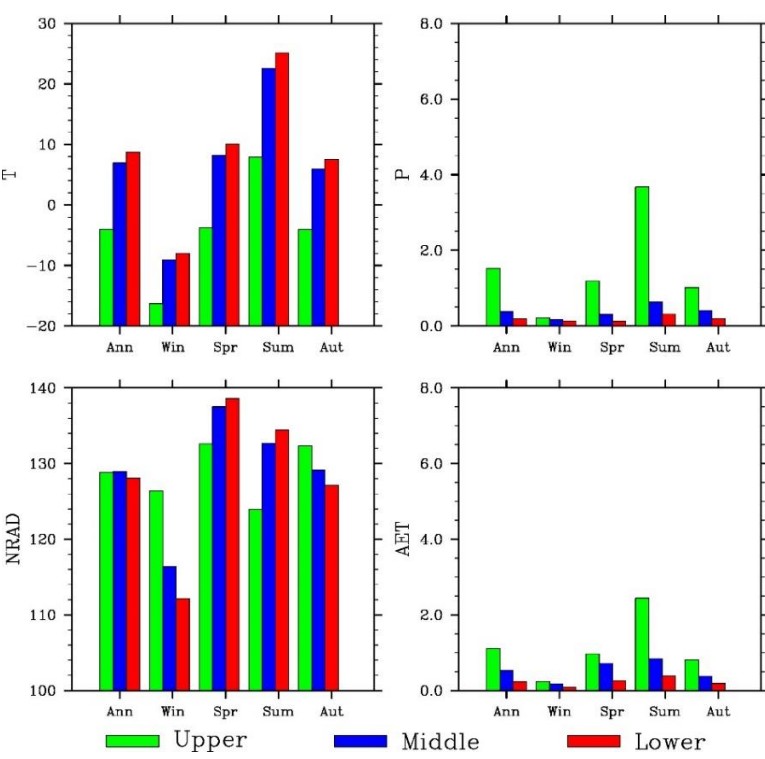


Figure 3. Seasonal variations of simulated air temperature ($T$, $^{o}$C), precipitation ($P$, mm/d), net
radiation (NRAD, W/m$^2$) and actual evapotranspiration (*AET*, mm/d) in three basin reaches
averaged over 1980-2010.

The inter-annual variability of regional $T$ and $P$ is similar between the middle and lower
basin reaches (Fig. 4). A few dry years (e.g., 1990, 2001, and 2008) and wet years (e.g., 1981,
1989, 2002, and 2007) can be found. The amplitude of variability is larger for $P$ than $T$,
especially in the upper basin. The variability of $AET$ is also similar between the lower and
middle basin reaches, but it differs from that in the upper basin during some periods (e.g.,
around 1985). The differences in $AET$ between the middle and upper basins are much smaller
in the magnitude than those for the meteorological properties.

The above features of close values and similar inter-annual variability in the simulated $T$
and $P$ between the middle and lower basin reaches are also seen in the observations (Fig. 4).
The simulated $T$ in all basin regions and $P$ in the middle and lower basin reaches are close to
the observed ones. However, the simulated $P$ is about 0.4 mm/d higher (about 1.6 mm/d for
simulation vs. 1.2 mm/d for observation). The weather site in the upper basin is located in
relatively flat and low valley, while the simulation grids have many points at high elevations
where $P$ is larger than at the valley locations. NRAD values have large inter-annual variability   with
little difference among the regions.

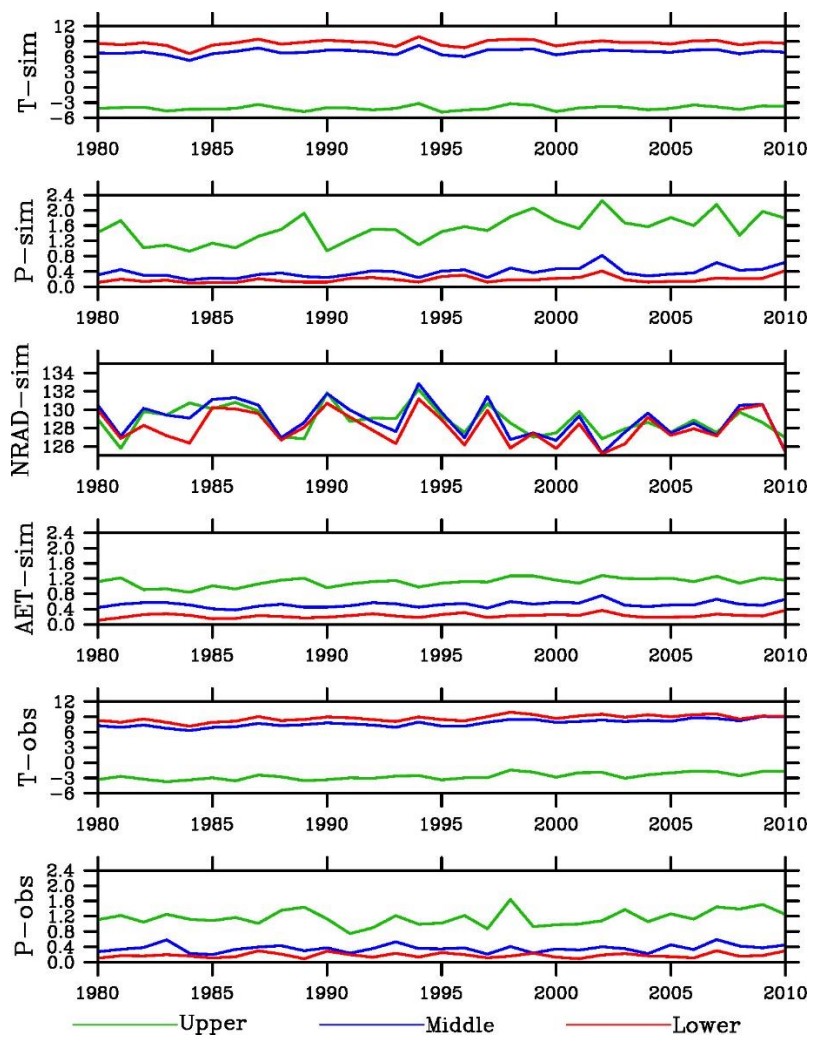

Figure 4. Inter-annual variations of simulated air temperature (*T*, °C), precipitation (*P*, mm/d),
net radiation (NRAD, W/m$^2$) and actual evapotranspiration (*AET*, mm/d), and observed air
temperature (*T*, °C) and precipitation (*P*, mm/d) in three basin reaches over 1980-2010.

The SPI for 12-month timescale also shows generally similar inter-annual variations over
the analysis period between the simulated and observed precipitation in the three basins (Fig.
5). In the upper basin, for example, the observed wet spells occurred around 30, 50, 120, 230,
290, 340, and 360 months, while the dry spells occurred around 20, 30, 70, 100, 180, 200, 260,
and 300 months. The simulation reproduces most of the wet and dry spells. However, the
simulation is too wet during about 40-80 months and largely misses the dry events during 240-
260 months.

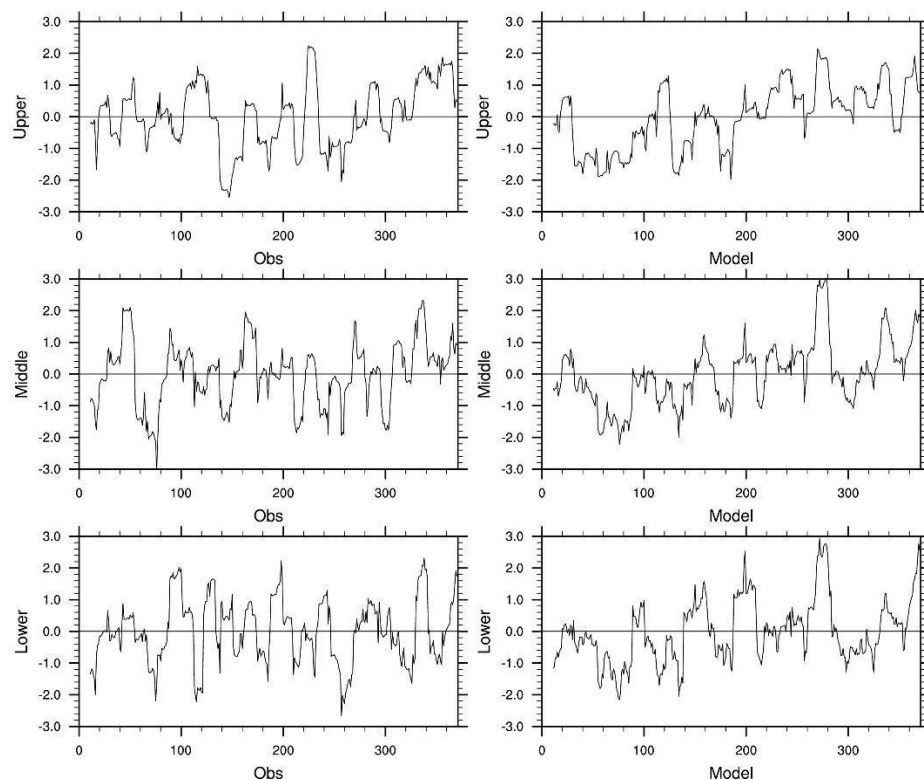

Figure 5. The Standardized Precipitation Index (SPI) for 12-month timescale over the analysis

period. The left and right are observation and simulation. From top to bottom are the upper, middle, and lower basins, respectively. The horizontal number is month from the beginning of the analysis period.

The simulated $P$ increases around 50% over the simulation period, statistically significant

at p<0.01in all basin reaches (Table 2). The simulated $AET$ also increases, but at a smaller

degree of around 20% and p<0.01 only in the upper basin. The simulated $T$ shows increasing

trends, but insignificant in all reaches. The simulated $P$ trends are close to the observed  ones

in the middle and lower basin reaches, but opposite to that in the upper basin. The simulated $T$

underestimates the observed warming, which was about 2°C at p<0.01.

Table 2. Mann-Kendall trends from 1980 to 2010 of simulated temperature ($T$), precipitation
($P$), and actual evapotranspiration ($AET$) and observed temperature ($T_{obs}$), precipitation ($P_{obs}$).
The bold and italic numbers are significant at $p<0.01$ and $p<0.05$, respectively.

| Variable | Upper | Middle | Lower |
|---|---|---|---|
| T(°C) | 0.4 | 0.4 | 0.4 |
| P (%) | **53.0** | **63.7** | **47.9** |
| AET (%) | **21.4** | 16.6 | 27.1 |
| $T_{obs}$ (°C) | **1.9** | **2.0** | 0.7 |
| $P_{obs}$ (%) | -10.7 | *74.6* | 62.5 |


## 3.2 Spatial patterns of aridity indices


*PET* calculated using the Penman-Monteith method is mostly 1.7-2.25 mm/d in the upper basin
(Fig. 6). It increases to above 3 mm/d in the middle and lower basins. There is little difference
between the two regions. The meteorological aridity index, *AI*, shows a similar pattern but
opposite gradient (Fig. 6). It mostly has a humid climate in the upper basin, but becomes mainly arid
climate in two other basin regions. The hydrological aridity index, *ESI*, has the same gradient as *AI*,
but with different spatial pattern (Fig. 6). It also mostly has a humid climate in the upper basin
and arid climate in the lower basin.  However, it is largely semi-arid climate in the middle basin. P
and AET are the highest in the upper basin and the lowest in the lower basin, while T and          PET
have an opposite seasonal cycle. This explains why AI and ESI are larger in the upper basin than the
middle or lower basin.

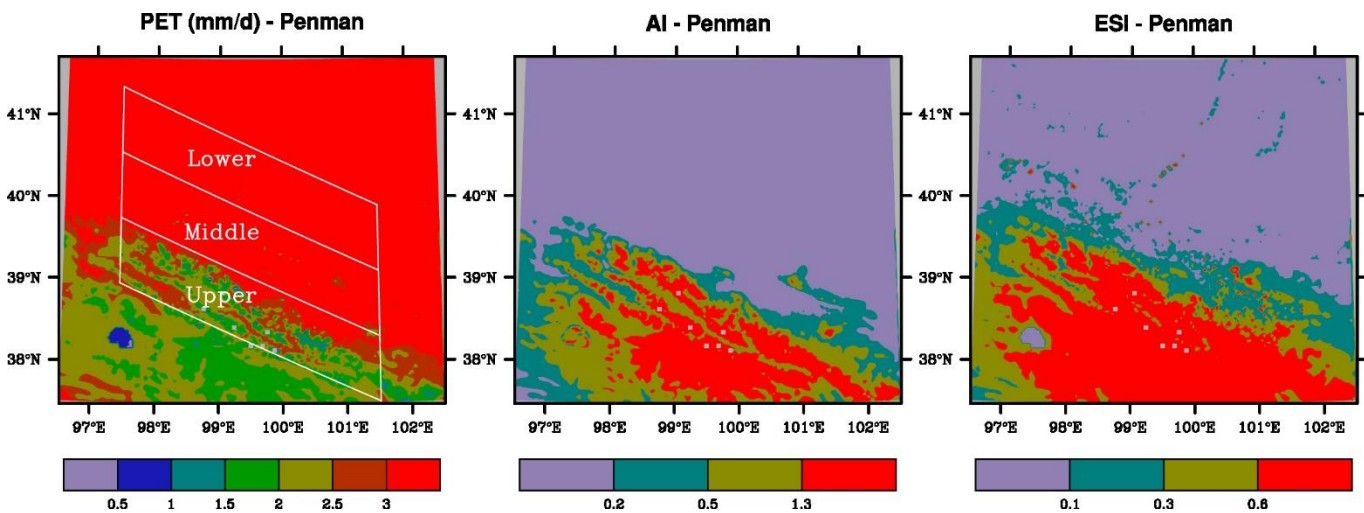

Figure 6. Spatial distributions of potential evaporation (*PET*, mm/d), Aridity index (*AI*) and
Evaporative Stress Index (*ESI*) with *PET* estimated using the Penman-Monteith method.
Averaged over 1980-2010. The Heihe River basins are shown in the left panel. The color bars from
left to right for AI and ESI are arid, semi-arid, semi-humid and humid climate.
*PET* calculated using the Hamon method has the same pattern as the one using the Penman-
Monteith method, but with smaller magnitude (Fig. 7). *PET* is mostly about 1 mm/d in the
upper basin and increases to about 1.5-1.75 mm/d in the middle basin, and further to 1.75-2.25
mm/d in the lower basin.

The different spatial patterns between *AI* and *ESI* seen above are also found for the Homan
method. *AI* is mostly above 0.6 in the upper basin (Fig. 7). It is below 0.2 in the middle and
lower basins without apparent differences between the two regions. In contrast, while *ESI*
remains large values of mostly above 0.9 in the upper basin and low values of below 0.2 in the
lower basin, the values in many areas of the middle basin are 0.4-0.9, much different from
those in the lower basin (Fig. 7).


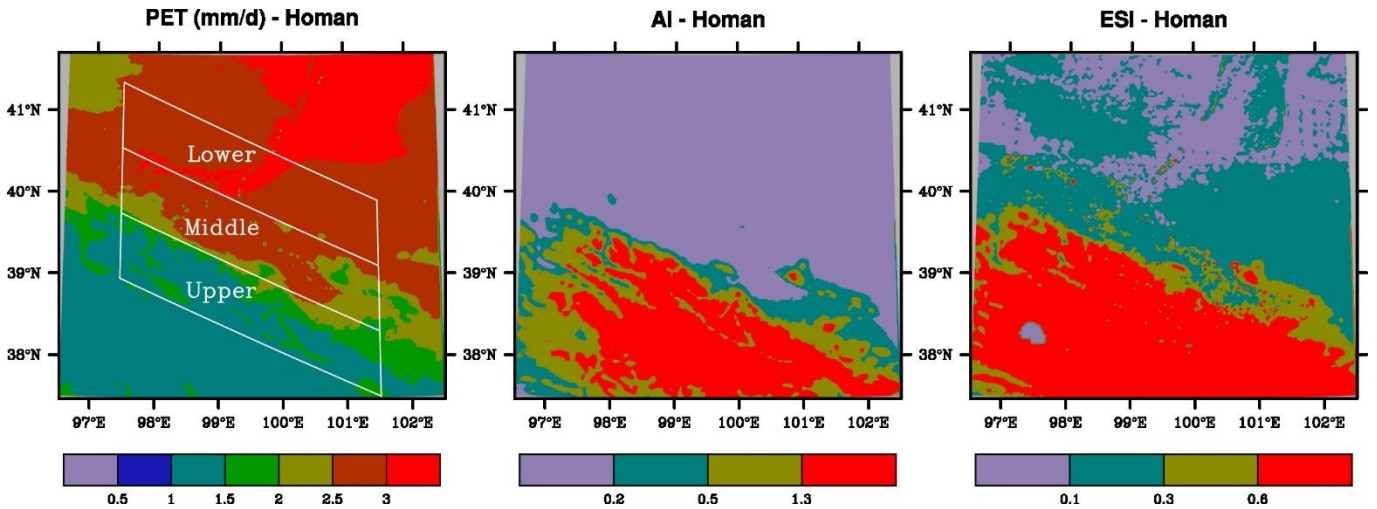

Figure 7. Spatial distributions of potential evaporation (*PET*, mm/d), Aridity index (*AI*) and
Evaporative Stress Index (*ESI*) with *PET* estimated using the Hamon method. Averaged over
1980-2010. The Heihe River basins are shown in the left panel. The color bars from left to right for AI
and ESI are arid, semi-arid, semi-humid and humid climate.

## 3.3 Climate classification

The annual *PET* averages over 1980-2010 calculated using the Penman method are 2.12, 3.91,
and 4.76 (Table 3 and Fig. 8). The corresponding *AI* values are about 0.9, 0.12, and 0.04, falling
into semi-humid, arid, and arid climate. The corresponding *ESI* values are 0.63, 0.22, and 0.07,
falling into humid, semi-arid, and arid climate. The annual *PET* averaged over 1980-2010
calculated using the Homan method are 1.25, 2.33, and 2.65 mm/d for the upper, middle, and
lower basin reaches. The corresponding *AI* values are about 1.3, 0.18, and 0.07, falling into
humid, arid, and arid climate. The corresponding *ESI* values are 0.78, 0.31, and 0.13, falling
into humid, semi-humid, and semi-arid climate. The averages of PET or each of the aridity index are
statistically significant ($p<0.01$) between any two regions of the Height River Basin.

Thus, the climate across the HRB classified using *AI* has two types of semi-humid (the
Penman method for *PET*) or humid (the Homan method) in the upper basin, and arid in both
middle and lower basin reaches. In contrast, the climate classified using *ESI* has three types of
humid in the upper basin, semi-arid (the Penman method) or semi-humid (the Homan method)
in the middle basin, and arid (the Penman method) or semi-arid (the Homan method) in the
lower basin. This indicates that only the hydrological aridity index is able to identify the
transition climate zone in the middle basin.

The difference between *AI* and *ESI* in classifying climate is related to the similar feature
with the meteorological variables. Annual *P* is 555 mm in the upper basin, which is
substantially different from 69-139 mm in the middle and lower basins. The mean *T* is -4.0$^o$C
in the upper basin, which is well below 6.9-8.7$^o$C in the middle and lower basin reaches. The
corresponding *PET* values fall into two groups, 299 mm in the upper basin and 672-767 mm
in the middle and lower basin reaches. This explains why the *AI* falls into two groups. In
contrast, *AET* is 226, 161, and 80 mm, substantially different not only between the middle and
upper reaches but also between the middle and lower reaches. This explains why the *ESI* falls
into three groups.
Table 3. Regional average (AVE), standard deviation (SD), and coefficient of variation (CV)
for potential evapotranspiration (*PET*, mm/d), aridity index (*AI*), and evaporative stress index
(*ESI*). A, SA, SH, and H represent arid, semi-arid, semi-humid, and humid climate, respectively.

| PET | Basin | PET | | | AI | | | ESI | | |
|---|---|---|---|---|---|---|---|---|---|---|
| | | AVE | SD | CV | AVE | SD | CV | AVE | SD | CV |
| Penman-Monteith | Upper | 2.12 | 0.12 | 0.06 | 0.90 (SH) | 0.32 | 0.35 | 0.62 (H) | 0.07 | 0.11 |
| | Middle | 3.91 | 0.21 | 0.05 | 0.12 (A) | 0.06 | 0.50 | 0.22 (SA) | 0.06 | 0.26 |
| | Lower | 4.76 | 0.29 | 0.06 | 0.04 (A) | 0.03 | 0.64 | 0.07 (A) | 0.03 | 0.41 |
| Hamon | Upper | 1.25 | 0.04 | 0.03 | 1.30 (H) | 0.37 | 0.29 | 0.78 (H) | 0.05 | 0.07 |
| | Middle | 2.33 | 0.11 | 0.05 | 0.18 (A) | 0.08 | 0.43 | 0.31 (SH) | 0.06 | 0.19 |
| | Lower | 2.65 | 0.16 | 0.06 | 0.07 (A) | 0.04 | 0.56 | 0.13 (SA) | 0.04 | 0.31 |

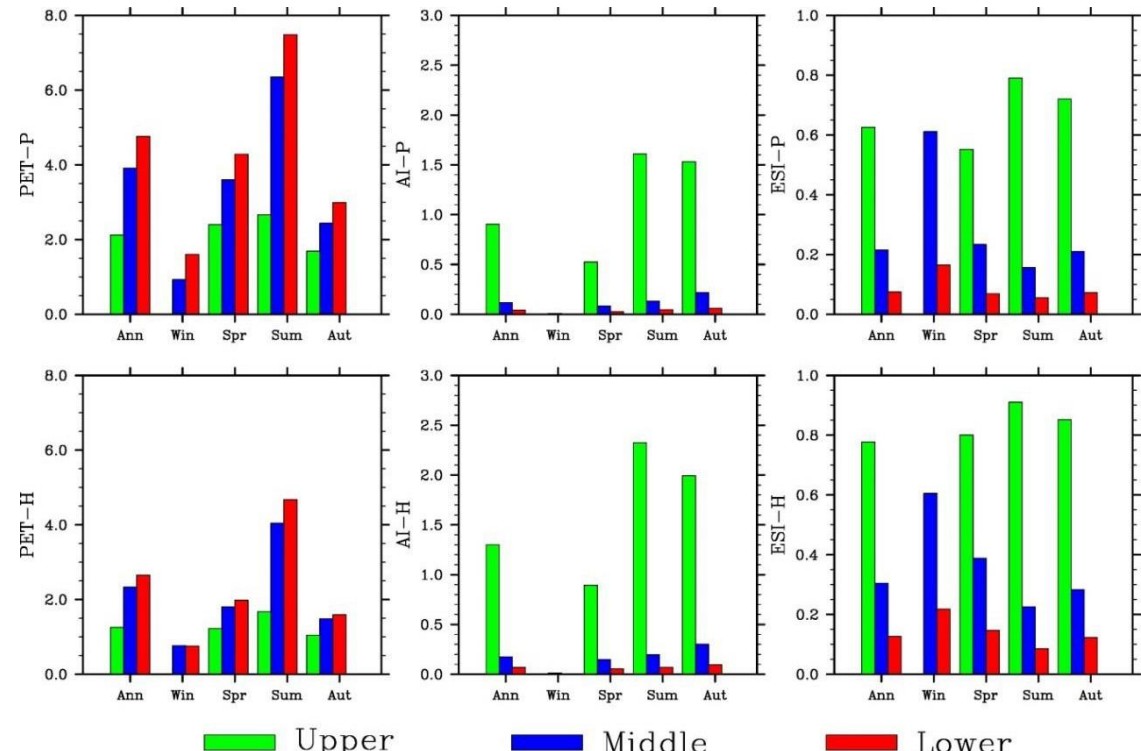

Figure 8. Seasonal variations of simulated potential evapotranspiration (*PET*, mm/d), Aridity
Index (*AI*), and Evaporative Stress Index (*ESI*) (from left to right). The top and bottom panels
are for the Penman-Monteith and Hamon method, respectively.

## 3.4 Temporal variations of aridity indices

### 3.4.1 Seasonal cycle

For the Penman-Monteith method, *PET* is the highest in summer and smallest in winter (Fig.
8). Note that winter PET in the upper basin is not shown because *T* is below zero on too many
days. The amplitude in the middle basin is close to that in the lower basin, but much larger
than that in the upper basin. Different from the upper basin where *AI* and *ESI* are also the
largest in summer, *AI* is the largest in fall, while *ESI* is the largest in winter in the middle basin
(as well as lower basin). The seasonal variations of *PET*, *AI* and *ESI* estimated using the
Homan method are similar to those using the Penman method.

The seasonal AI and ESI cycles are related to those of the meteorological and hydrological
conditions. *T*, *P* and *AET* (Fig. 3), and *PET* (Fig. 8) all increase from winter to summer. In the
upper basin, the increases in *P* and *AET* from spring / fall to summer are larger than the

corresponding increases in *PET*, leading to larger *AI* and *ESI* values in summer. In the middle as well
as lower basin, however, *PET* increases substantially from spring / fall, leading to
smaller *AI* and *ESI* in summer than in spring / fall.

**3.4.2 Inter-annual variability**

*PET* in the middle basin calculated using the Penman-Monteith method shows similar inter-
annual variability over the period of 1980-2010 to that in the lower basin, but much different
from that in the upper basin (Fig. 9). The standard deviation (SD) increases from the upper
(0.12) to middle (0.21) and to lower basin (0.29) (Table 2). The coefficient of variation (CV)
(the ratio of the standard deviation to the average), a statistical property often used to measure
relative variability intensity, however, is comparative among the reaches.

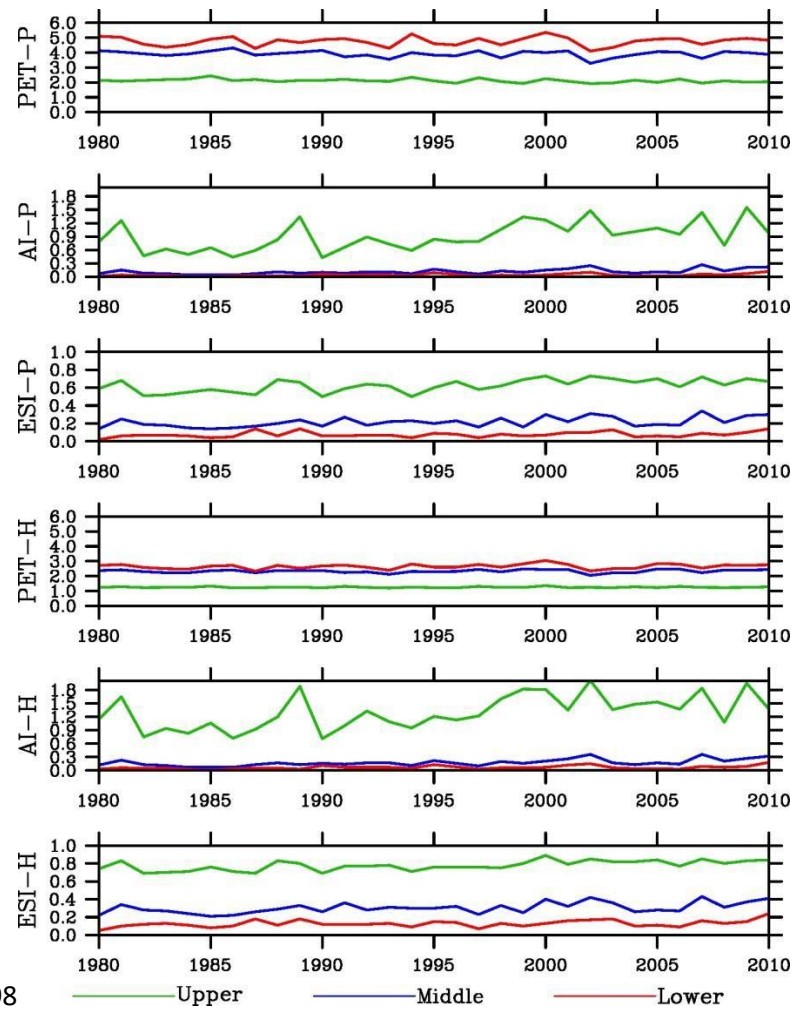

298

Figure 9. Inter-annual variations of potential evapotranspiration (*PET*, mm/d), Aridity Index (*AI*), and Evaporative Stress Index (*ESI*). P and H indicates the Penman-Monteith and Hamon method, respectively.

The SD values of both *AI* and *ESI* decrease from the upper to middle and to lower basin. However, SD of *AI* (*ESI*) in the middle basin is much closer to that in the lower (upper) basin. The CV values have opposite gradient to SD, increasing from the upper to middle and to lower basin. In addition, CV differs mainly not between the basin reaches but between aridity indices: *AI* is larger than *ESI*.

### 3.4.3 Long-term trends

*PET* shows little trends over the simulation period (Table 4). In contrast, aridity indices increased dramatically, by 60% or more for *AI* and 15-50% for *ESI*. The trends are significant at $p < 0.01$ in the upper and middle basin reaches and $p < 0.05$ in the lower basin. The results indicate a less dryness condition in the HRB, which is the more remarkable in the middle than upper basin and in the meteorological than hydrological aridity index. Increase in precipitation is a major contributor.

Table 4. Mann-Kendall trends from 1980 to 2010 of potential evapotranspiration (*PET*), Aridity Index (*AI*), and Evaporative Stress Index (*ESI*) (in%). *P* (*H*) indicates the Penman-Monteith (Hamon) method. The bold and italic numbers are significant at $p < 0.01$ and $p < 0.05$.

| Index | Upper | Middle | Lower |
|-------|-------|--------|-------|
| PET-P | *-7.3* | -2.7 | 0.3 |
| AI-P | **72.5** | **98.6** | *80.9* |
| ESI-P | **24.8** | **51.4** | *47.8* |
| PET-H | 0.0 | 2.7 | 3.6 |
| AI-H | **62.6** | **84.3** | *66.3* |
| ESI-H | **16.2** | **40.8** | *40.5* |

### 3.5 Extreme events

The aridity indices for 4 simulated dry years (1982, 1990, 2001, and 2008) and 4 wet years (1981, 1989, 2002, and 2007) (Figs.10-11) and the averages over the dry or wet years (Fig. 12) were analyzed. The annual *AI* values using the Penman-Monteith method are 0.4-0.5 for the

first two dry years and 0.7-1.0 for the last two years in the upper valley (Fig. 12). The average
over the 4 years is about 0.65. In comparison, the average is about 0.9 over 1980-2010 and 1.4
over the 4 wet years. The values are very small in spring (except in 1982) and occasionally in
fall (1990). The annual *AI* values in the middle and lower basin reaches are below 0.2 for
individual dry years and average. The small values are found for individual seasons  except
falls of the last two years in the middle basin. In compassion, the annual values are 0.4 or above
in 3 falls of the 4 wet years.

The annual *ESI* values using the Penman-Monteith method are 0.5 or larger in the upper
valley. The average over the 4 years are nearly 0.6. In comparison, the average is about 0.62
over 1980-2010 and 0.7 over the 4 wet years. The values are comparable from spring to fall,
though relatively smaller in spring. This is different from AI. The annual *ESI* values are about
0.2 in the middle and below 0.1 in the lower basin for individual dry years and average. Thus,
the values are apparently different between the middle and lower basin reaches. This is another
difference from AI. The lowest values mostly occur in summer in both basin reaches. In
compassion, the annual values are 0.25-0.35 in the middle basin and 0.1 or larger in 3 of the 4
wet years in the lower basin.

Same results can be found for the Hamon method, that is, substantially smaller *AI* than
normal, especially in spring but no much *ESI* changes from normal and between seasons in the
upper basin, and no much *AI* change from normal and wet events (small in all cases) in the
middle and lower basin reaches but much smaller *ESI* than wet events and different between
the two basin reaches, though slightly larger *AI* and *ESI* values. The results suggest that *ESI* is
better representative of extreme dry conditions in the middle basin, but less sensitive to aridity
in the upper basin.

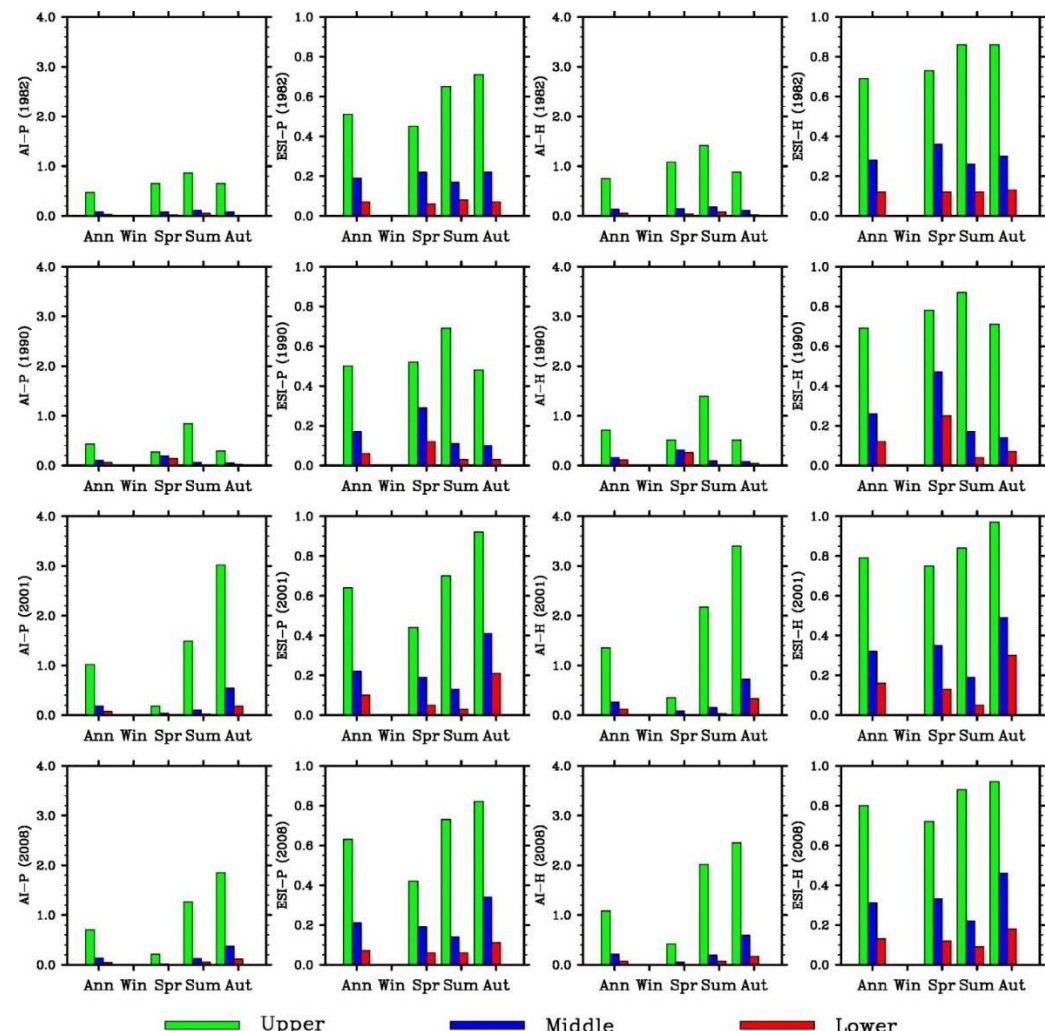

Figure 10. Seasonal variations of simulated Aridity Index (*AI*), and Evaporative Stress Index (*ESI*) using the Penman-Monteith and Hamon methods (left to right) for the dry years of 1982,1990, 2001, and 2008 (from top to bottom).

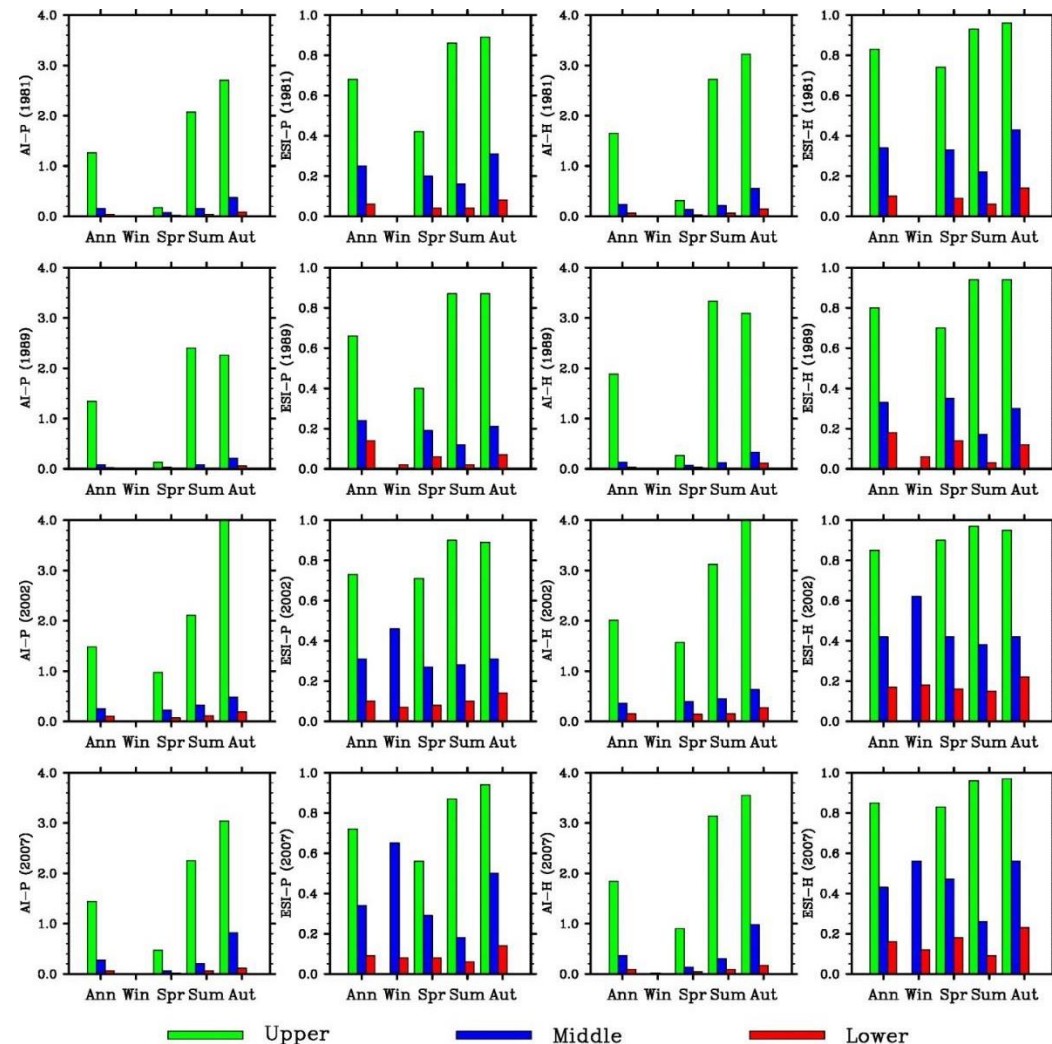

Figure 11. Seasonal variations of simulated Aridity Index (*AI*), and Evaporative Stress Index

(*ESI*) using the Penman-Monteith and Hamon methods (left to right) for the wet years of 1981,

1989, 2002, and 2007 (from top to bottom).

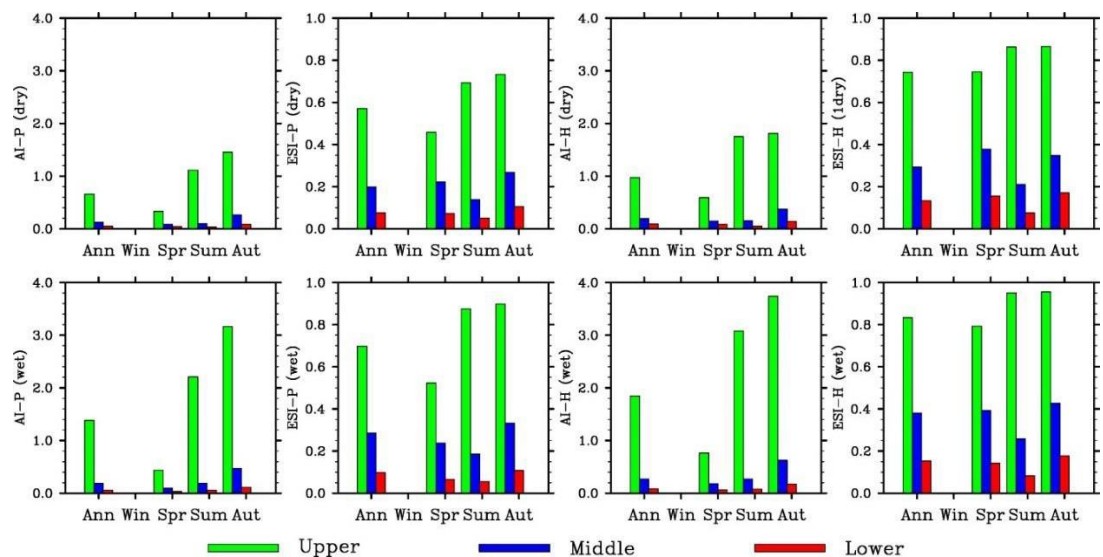

Figure 12. Seasonal variations of simulated Aridity Index (*AI*), and Evaporative Stress Index (*ESI*) using the Penman-Monteith and Hamon methods (left to right) for averages over the dry years of 1982, 1990, 2001, 2008 (top) and (bottom).

**4 Discussion**

**4.1 Supports to the integrated water–ecosystem–economy study in the HRB**

The HRB is a typical inland river basin with a strong contrast in topography, landscape, climate, and human activities from the headwater to end point along its drainage system. Comprehensive monitoring, modeling, and data manipulation studies have been conducted for several decades to understand the hydrological and ecological processes and interactions in the HRB (Cheng et al., 2014). The middle HRB is a special region with dynamic land cover and use changes due to human activity. Different from the upper HRB regions where climate change has been the controlling factor for hydrological and ecological processes, surface water condition is extremely important in the middle HRB where irrigated farmland is the largest land use and natural oases have been gradually replaced by artificial oases (Li et al., 2001, Cheng et al., 2014). According to our study, hydrological index ESI should be a better indictor than the meteorological index AI for water supply and demand conditions in the middle HRB. Zhang et al. (2014) found that the streamflow from the upper to middle HRB has risen due to climate change, but the streamflow from middle to lower HRB has reduced. They attributed this reduction to increasing water consumption by human activities in the middle HRB. Our study indicates less dryness trend in the middle HRB and therefore supports the analysis that climate change was not a major factor for the reduction. Sun et al. (2015) found an increasing

trend in vegetation growth in the middle HRB and attributed it to irrigation. Our study  shows
less drying trend in this region, suggesting that more net water was another contributor to  the
increasing vegetation growth.

**4.2 Importance of land-surface processes**

The water shortage and frequent droughts are the biggest environmental threat to the
ecosystems and human activities in the HRB as well as entire northwestern China. This
comparison study provides evidence for the importance of water and energy interactions
between land process and the atmosphere and between upstream and downstream in
determining climate types in an arid climate. Because the *ESI* values are related to *AET* that is
controlled by land-surface properties and management practices (e.g., rainfall-fed crops  vs
irrigated crops; natural wetlands vs cultivated drained croplands), our results suggest the land-
surface processes play an important role in affecting aridity conditions. The landscape in the
HRB, especially its transition zone, has changed remarkably in the past several decades due to
urbanization, farming, and grazing activities (Hu et al., 2015). The irrigation may have caused
the lower basin more water stressed (higher *ESI* than *AI*) since stream water from Heihe  is
intercepted and rivers go dry downstream. The *ESI* should reflect this change since it is
calculated partially based on the land-surface hydrological conditions. Urbanization, farming,
and grazing would reduce vegetation coverage. This would further reduce  evapotranspiration
and increase runoff. Irrigation would play opposite roles. The RIEMS model uses the
Biosphere and Atmosphere Transfer Scheme (BATS) (Dickinson and Henderson-Sellers, 1993)
to simulate the land-surface hydrological processes. The vegetation and soil properties
measured in the HRB in 2000 were used to replace the universal BATS specifications,  which
improved precipitation simulation (Xiong and Yan, 2013). However, the above disturbance
over time were not included in the simulation that provided the data for this study. Numerical
experiments with this model are needed to provide quantitative evidence for the  hydrological
effects of the disturbances.

**4.3 Role in moderating climate**
The magnitude of *AI* (*ESI*) inter-annual variability in the middle basin is (is not) very close to
that in the lower basin, another evidence for the unique capacity of *ESI* in separating the climate
zones between the middle and lower basin reaches. The magnitude of the relative inter-annual
variability differs mainly between *AI* and *ESI*, larger with *AI*. In addition, both *AI* and *ESI* in
the HRB decreased dramatically from 1980 to 2010, at greater rate with *AI*. Thus, the aridity
conditions described using *ESI* is less variable, suggesting the role of local hydrological
processes in moderating extreme climate events.

**4.4 Future trends**

One of the hydrological consequences from the projected climate change due to the greenhouse
gas increase is more frequent and intense droughts in watersheds of dry regions. In the
Colorado River Basin, global warming may lead to substantial water supply shortages
(McCabe and Wolock, 2007), and the climate models projected considerably more drought
activities in the 21$^{st}$ century (Cayan et al., 2010). In the HRB, the climate of the upper HRB
will likely become warmer and wetter in the near future (Zhang et al., 2016), consistent with
the historical records. Correspondingly the basin-wide evapotranspiration, snowmelt, and
runoff are projected to increase over the same period. Many aridity indices, including the *AI*,
have been used to project future aridity trends (Paulo et al., 2012). However, most of the recent
ESI studies are based on historical remote sensing for monitoring short-term drought
development, which limits the application of this index to climate change impact research. Due
to the unique ability with the *ESI* in identifying the transition climate zone as shown in this
study, it would be valuable to explore its potential for future aridity projection study and
compare with that of the *AI*.

**4.5 Uncertainty and future research**

The regional climate simulation which generated data for this analysis has many uncertainties
(Xiong and Yan, 2013). One of the contributing factors is the very limited number of
meteorological, hydrological, and ecological measurement sites. A large-scale, multiple-year
field experiment project has been conducted in the HRB, which have been generating extensive
datasets (Wang et al., 2014). These data are being used to improve the regional climate

modeling, which will in turn generate new high-resolution data for further aridity analysis. Furthermore, the regional climate modeling has been expanded into the middle 21$^{st}$ century, providing data for calculating the aridity indices and comparing their future trends. Comparisons of other meteorological and hydrological aridity indices are also a future research issue.

**5 Conclusions**

This study has found that the *ESI* climate classification agrees with the Koppen climate classification (Peel et al., 2007). By using *ESI*, we found that the climate types are different among the upper, middle, and lower HRB. In contrast, there is no difference between the middle and lower HRB regions when the *AI* is used. The comparison results from this study therefore suggest that only ESI is able to identify a transition climate zone between the relatively humid climate in the mountains and the arid climate in the Gobi desert region. We conclude that the hydrological aridity index *ESI* is a better index than the meteorological aridity index *AI* for climate classification in the HRB with a complex topography and land cover. Selection of the most appropriate aridity index facilitates climate characterization and assessment, risk mitigation, and water resources management in the arid region.

**Acknowledgement** This study was supported by the National Natural Science Foundation of China (NSFC) (No. 91425301) and the USDA Forest Service. We thank the reviewers for valuable and insightful comments and suggestions.

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
