# Peer review of "Identifying a Transition Climate Zone in an Arid River Basin using Evaporative Stress"

_Natural Hazards and Earth System Sciences, 2017_

## Referee Comment (RC1) · Anonymous Referee #1 · 1 Mar 2018

Drought is the most severe natural disaster in the northwestern inland area of China. The research on drought is essential for scientific disciplines from geography to ecology. Drought research has traditionally focused on the regional features. However, there are new demands for more attentions to the local features due to the rapid changes in land cover and land use. This study addresses the demand by examining the capacity of drought indices in identifying the local climate regimes in a mountain region the northwestern China. It should provide valuable information for the research of other scientific disciplines in the dry area.

I have a couple of suggestions that may improve this manuscript. First, a multiple-disciplinary comprehensive research project has been conducted in the Heihe River basin. Please discuss the implications of the findings for the research of other scientific disciplines in the study area. Secondly, there are many drought indices available. Please provide a brief review on these indices.

---

## Author Comment (AC1) · 12 Apr 2018

[revised manuscript text omitted]

The Koppen climate classification, one of the most widely used climate classification
techniques at large geographic scales and constructed based on the properties of ecosystems,
latitude, and average and seasonal precipitation and temperature, is often used for a large
region with static environmental conditions. Drought indices are another useful tool to
classify and monitor aridity and drought of a region.  Drought indices are quantitative
measures of drought levels by combining one or several variables (indicators) into a single
numerical value (Zargar et al., 2011). Drought indices are usually categorized into
meteorological, hydrological, agricultural, and social droughts (Wilhite and Glantz, 1985).
Different definitions can be found. The first two types of droughts investigated in this study
were simply considered as a lack of water due to anomalous atmospheric and land-surface
conditions, respectively.

Precipitation, temperature and humidity are atmospheric conditions often used to estimate
meteorological drought indices. Percent of Normal (PN) and Standardized Precipitation
Index (SPI) (McKee et al., 1993) are simply based on precipitation and can be used to
measure anomalies of a period over various lengths.  Palmer Drought Severity Index (PDSI)
(Palmer, 1965) and Keetch-Byram Index model (Keetch and Byram, 1968) are based on
water supply and demand estimated mainly using precipitation and temperature (Guttman,
1999).  Both indices depend on precedent daily or monthly values, making them specifically
useful for a persistent event like drought. The aridity index (AI) (Budyko, 1974) uses annual
averages of precipitation and potential evapotranspiration (PET), which is mainly determined
by temperature. AI is also used as an essential element in many other indices to describe
actual drought conditions (Arora, 2002). Different from the above four indices, which are often used to indicate the dry spells of temporal humidity variability, AI is often used to
indicate a climatic humidity condition of a region.

[revised manuscript text omitted]

---

## Referee Comment (RC2) · Anonymous Referee #2 · 8 Jun 2018

In the manuscript entitled "Identifying a Transition Climate Zone in an Arid River Basin using a Hydrological Drought Index", Zhang et. al., analyzed meteorological and hydrological drought indices to identify transitional climate zone in a Heihe River Basin (HRB) in the arid northwestern China. The authors used simulations from a Regional Integrated Environmental Model System (RIEMS 2.0). Based on their analyses, the authors found the hydrologic based drought index being more suitable for the characterizing the transitional climate zone in the study basin, compared to meteorological based drought index. While the study might fall within the scope of this Journal, there are several limitations, in my opinion the work performed here is not adequate for the publication in this Journal. Major issues with this study is detailed below.

1. I find the narration as authors put up that drought indices are used for climate

classification a bit strange. The aridity index as used by the authors as drought index is also strange as this is generally used as climate characteristics (starting from the Budyko's classical work). Also what I have hard time to understand is that the authors are using the full range of aridity index – which considers both wet and dry phase – in their analyses. So where is the perspective of droughts here? In order to consider droughts the authors must focus of one end of the drought indices (drier parts) and not the entire range. The similar is the case with the author's analyses for the hydrological drought index. So, if I did not get it wrong – the whole perspective of the author's analyses revolving around the drought indices are misleading.

2. The authors must provide argument and reasoning for the choice of selected drought indices. Why do not authors choose conventional and commonly used drought index – Standardized precipitation index (SPI) or the Standardized Precipitation-Evapotranspiration Index (SPEI) for meteorological droughts; and runoff based drought index for hydrological droughts.

3. Related to the above one, why are the drought indices analyzed at annual time scale and not monthly or moving average estimates as commonly used in drought studies?

4. To a certain degree I understand the choice of using the regional climate model in their analyses, but what I miss is the thorough comparison of the RIEMS model to observations – in the sense that it could provide meaning conclusion for drought analysis. The authors must demonstrate that the selected model is able to capture the observed behavior of meteorological and hydrological droughts (say SPI or standardized runoff index). Here I mean skill of the model for drought index and not the variable itself.

5. To my understanding the RIEMS model provide estimates of net radiation (Rn) through their numerical parameterization of the mass, momentum and energy conservations schemes – so why do not the authors use Rn variable instead of PET – which is just the proxy of available energy? Please elaborate.

6. As stated by the Abstract and Discussion sections point towards "human disturbance" affecting the hydrological drought conditions. Do these statements are supported by their analyses or this is just the speculation? Do the selected model (RIEMS) considers the process affected by human disturbances (irrigation, farming, urbanization, grazing activities) and in which way those disturbances affect the hydrological processes? Please consider elaborating on the RIEMS parameterization and provide some modeling results in this direction.

7. All you describe in Section 3.2 is hydro-climatic characteristics and not the spatial pattern of drought indices – as the section heading indicates and explained in text. See the point 1 of my comment.

8. The starting paragraph in the Introduction section does not flow - instead of making a case (motivation) for the current study, it starts with describing the study catchment (Heihe river) and then jumping to other region (Colorado river in US). Please reformulate.

9. The results section starts with detailing supplement plots – I would have expected to see first the main plots and then the supporting plots and not other way around.

10. Figure 1: Quality is too poor – I could not read any of the subplot's legend.

11. Figures 2-7: What is the point of making these figures up to 43oN, when the whole study area ends at 42oN? Also please consider improving these figures and limit them to just show the study basin region.

---

## Author Response (AR1)

| 1 | **Review comments** |
| 2 | |
| 3 | **Anonymous Referee #21** |
| 4 | |
| 5 | Drought is the most severe natural disaster in the northwestern inland area of China. The |
| 6 | research on drought is essential for scientific disciplines from geography to ecology. Drought |
| 7 | research has traditionally focused on the regional features. However, there are new demands |
| 8 | for more attentions to the local features due to the rapid changes in land cover and land use. |
| 9 | This study addresses the demand by examining the capacity of drought indices in identifying |
| 10 | the local climate regimes in a mountain region the northwestern China. It should provide |
| 11 | valuable information for the research of other scientific disciplines in the dry area. |
| 12 | |
| 13 | I have a couple of suggestions that may improve this manuscript. First, a multiple-disciplinary |
| 14 | comprehensive research project has been conducted in the Heihe River basin. Please discuss |
| 15 | the implications of the findings for the research of other scientific disciplines in the study area. |
| 16 | Secondly, there are many drought indices available. Please provide a brief review on these |
| 17 | indices. |
| 18 | |
| 19 | **Anonymous Referee #2** |
| 20 | |
| 21 | In the manuscript entitled "Identifying a Transition Climate Zone in an Arid River Basin |
| 22 | using a Hydrological Drought Index", Zhang et. al., analyzed meteorological and |
| 23 | hydrological |
| 24 | drought indices to identify transitional climate zone in a Heihe River Basin |
| 25 | (HRB) in the arid northwestern China. The authors used simulations from a Regional |
| 26 | Integrated Environmental Model System (RIEMS 2.0). Based on their analyses, the |
| 27 | authors found the hydrologic based drought index being more suitable for the characterizing |
| 28 | the transitional climate zone in the study basin, compared to meteorological |
| 29 | based drought index. While the study might fall within the scope of this Journal, there |
| 30 | are several limitations, in my opinion the work performed here is not adequate for the |
| 31 | publication in this Journal. Major issues with this study is detailed below. |

1. I find the narration as authors put up that drought indices are used for climate classification a bit strange. The aridity index as used by the authors as drought index is also strange as this is generally used as climate characteristics (starting from the Budyko's classical work). Also what I have hard time to understand is that the authors are using the full range of aridity index – which considers both wet and dry phase – in their analyses. So where is the perspective of droughts here? In order to consider droughts the authors must focus of one end of the drought indices (drier parts) and not the entire range. The similar is the case with the author's analyses for the hydrological drought index. So, if I did not get it wrong – the whole perspective of the author's analyses revolving around the drought indices are misleading.

2. The authors must provide argument and reasoning for the choice of selected drought indices. Why do not authors choose conventional and commonly used drought index – Standardized precipitation index (SPI) or the Standardized Precipitation-Evapotranspiration Index (SPEI) for meteorological droughts; and runoff based drought index for hydrological droughts.

3. Related to the above one, why are the drought indices analyzed at annual time scale and not monthly or moving average estimates as commonly used in drought studies?

4. To a certain degree I understand the choice of using the regional climate model in their analyses, but what I miss is the thorough comparison of the RIEMS model to observations – in the sense that it could provide meaning conclusion for drought analysis. The authors must demonstrate that the selected model is able to capture the observed behavior of meteorological and hydrological droughts (say SPI or standardized runoff index). Here I mean skill of the model for drought index and not the variable itself.

5. To my understanding the RIEMS model provide estimates of net radiation (Rn) through their numerical parameterization of the mass, momentum and energy conservations schemes – so why do not the authors use Rn variable instead of PET – which is just the proxy of available energy? Please elaborate. bance" affecting the hydrological drought conditions. Do these statements are supported by their analyses or this is just the speculation? Do the selected model (RIEMS)

considers the process affected by human disturbances (irrigation, farming, urbanization, grazing activities) and in which way those disturbances affect the hydrological processes? Please consider elaborating on the RIEMS parameterization and provide some modeling results in this direction.

7. All you describe in Section 3.2 is hydro-climatic characteristics and not the spatial pattern of drought indices – as the section heading indicates and explained in text. See the point 1 of my comment.

8. The starting paragraph in the Introduction section does not flow - instead of making a case (motivation) for the current study, it starts with describing the study catchment (Heihe river) and then jumping to other region (Colorado river in US). Please reformulate.

9. The results section starts with detailing supplement plots – I would have expected to see first the main plots and then the supporting plots and not other way around.

10. Figure 1: Quality is too poor – I could not read any of the subplot's legend.

11. Figures 2-7: What is the point of making these figures up to 43oN, when the whole study area ends at 42oN? Also please consider improving these figures and limit them to just show the study basin region.

Interactive comment on Nat. Hazards Earth Syst

**Responses to comments from Reviewer 1**

(Note: Responses are provided in *italic* font)

I have a couple of suggestions that may improve this manuscript. First, a multiple-disciplinary comprehensive research project has been conducted in the Heihe River basin. Please discuss the implications of the findings for the research of other scientific disciplines in the study area. Secondly, there are many drought indices available. Please provide a brief review on these indices.

*Thanks for the valuable and insightful comments and suggestions. Revisions are made accordingly.*

*(1) A paragraph is added to discuss the significance of this study for the research on the Heihe River Basin (L420-439). Several existing studies on comprehensive monitoring, modeling and data manipulation, land cover and land use changes, streamflow, and vegetation are described. Supporting evidence and /or different interpretations from our study are provided. The discussion is focused on the relationships and interactions between the transitional middle HRB and other regions.*

*(2) A description of drought indices is added in the introduction section (L 80-111). The meteorological and hydrological droughts are defined. Several major drought indices for each of the two types of droughts are briefly described. Comparisons are made to indicate the circumstances for each index.*

**Responses to comments from Reviewer 2**

In the manuscript entitled "Identifying a Transition Climate Zone in an Arid River Basin using a Hydrological Drought Index", Zhang et. al., analyzed meteorological and hydrological drought indices to identify transitional climate zone in a Heihe River Basin (HRB) in the arid northwestern China. The authors used simulations from a Regional Integrated Environmental Model System (RIEMS 2.0). Based on their analyses, the authors found the hydrologic based drought index being more suitable for the characterizing the transitional climate zone in the study basin, compared to meteorological based drought index. While the study might fall within the scope of this Journal, there are several limitations, in my opinion the work performed here is not adequate for the publication in this Journal. Major issues with this study is detailed below.

*Thanks for reviewing our manuscript and providing many constructive and valuable comments and suggestions. We agree with the major concerns and confusions raised by the reviewer. We think that they are caused by our misuse of the drought index terms rather than by using inappropriate indices. In fact, we actually used correct indicators of aridity indices for this climate classification study. For this reason, we think we are able to address the comments 1-3 and 7 and by using aridity indices to replace drought indices and providing clarification. As suggested, SPI analysis and a brief description*

*of model parameterization are provided. They should address the comments 4-6. Some writings of*
*contexts or the way to present them are changed to address the comments 6-9. Figures are improved*
*to address the comments 10-11.*
*Please see below for detailed responses.*
2. I find the narration as authors put up that drought indices are used for climate
classification a bit strange. The aridity index as used by the authors as drought index
is also strange as this is generally used as climate characteristics (starting from the
Budyko's classical work). Also what I have hard time to understand is that the authors
are using the full range of aridity index – which considers both wet and dry phase –
in their analyses. So where is the perspective of droughts here? In order to consider
droughts the authors must focus of one end of the drought indices (drier parts) and not
the entire range. The similar is the case with the author's analyses for the hydrological
drought index. So, if I did not get it wrong – the whole perspective of the author's
analyses revolving around the drought indices are misleading
*It is a very valuable and helpful comment. This study is to classify climate regimes in the*
*Heihe River watershed of the arid northwestern China. As pointed by the reviewer and also*
*indicated in other literatures (https://en.wikipedia.org/wiki/Climate_classificationin), aridity*
*indices are one of the indicators used in climate classification. One of the two indices used in*
*this study, the Budyko-type Aridity Index (AI), is apparently an aridity index. The difference*
*between potential (PET) and actual evaporation (AET) is often used to measure aridity*
*(https://www.springer.com/us/book/9783642291036, P 21-39). Thus, the other index used in*
*this study, the Evaporative Stress Index (ESI), is also an aridity index, though it appears as a*
*ratio of AET to PET instead of a difference between PET and AET. However, we misused*
*terms by calling the two aridity indices as drought indices and therefore caused the*
*confusions. In the revision, the terms are changed from meteorological and hydrological*
*drought indices to meteorological and hydrological aridity indices throughout the paper.*
*As clarified above, this study is to classify climate regimes rather than analyze droughts.*
*Besides the arid lower Heihe River reach, this region also includes the humid upper reach in*
*the mountain area and a transition zone in between the arid and humid regions. Thus, the full*
*range of aridity indices was used with lower and higher values indicating arid and humid*
*climates, respectively.*
2. The authors must provide argument and reasoning for the choice of selected
drought indices. Why do not authors choose conventional and commonly used drought
index – Standardized precipitation index (SPI) or the Standardized Precipitation-
Evapotranspiration Index (SPEI) for meteorological droughts; and runoff based drought
index for hydrological droughts.
*As indicated above, this study actually used aridity indices rather than drought indices for*
*climate classification. Considering many similarities between aridity and drought indices, we*
*describe and compare with some popular meteorological and hydrological drought indices*

*in the Introduction section. As suggested, SPI is analyzed in the revision (see the response to*
*comment 4).*

3. Related to the above one, why are the drought indices analyzed at annual time scale
and not monthly or moving average estimates as commonly used in drought studies?

*Due to the limitation with PET calculation in the upper basin, where no values are available*
*during much of the winter time when temperature is below $0^oC$, we did not analyze the*
*aridity indices at monthly time scale. However, we compared averages over the analysis*
*period for each of the spring, summer and fall seasons (Figure 8). Same as the annual*
*values, noticeable differences are found in ESI but not in AI between the middle and upper*
*basins for each of the three seasons.*

4. To a certain degree I understand the choice of using the regional climate model in
their analyses, but what I miss is the thorough comparison of the RIEMS model to
observations
– in the sense that it could provide meaning conclusion for drought analysis.
The authors must demonstrate that the selected model is able to capture the observed
behavior of meteorological and hydrological droughts (say SPI or standardized runoff
index). Here I mean skill of the model for drought index and not the variable itself.
*As suggested, SPI is analyzed (Figure 5). The results show general agreement between the simulated*
*and observed precipitation variability at basin reach scale. The results are described in the revision*
*(Lines 257-263).*

5. To my understanding the RIEMS model provide estimates of net radiation (Rn)
through their numerical parameterization of the mass, momentum and energy conservations
schemes – so why do not the authors use Rn variable instead of PET – which
is just the proxy of available energy? Please elaborate.

*PET is used because it is an item in the formulas to calculate the meteorological and*
*hydrological aridity indices AI (Line 151) and ESI (Line 163). It is true the Rn is produced by*
*the RIEMS model. Rn is used in calculating PET with the Penman-Monteith method (Eq.1).*

6. As stated by the Abstract and Discussion sections point towards "human disturbance"
affecting the hydrological drought conditions. Do these statements are supported by their
analyses or this is just the speculation? Do the selected model (RIEMS) considers the process
affected by human disturbances (irrigation, farming, urbanization, grazing activities) and in
which way those disturbances affect the hydrological processes? Please consider elaborating
on the RIEMS parameterization and provide some modeling results in this direction.

*This study did not provide direct results on the effects of the land-surface processes and*
*human disturbance on hydrological aridity conditions. Thus, the statements are just*
*speculations. For this reason, this statement is removed from the abstract.*

*The RIEMS model used the Biosphere and Atmosphere Transfer Scheme (BATS) to simulate*
*the land-surface hydrological processes. These processes depend on vegetation and soil*

*properties, which change over time due to the human and natural disturbances. The*
*vegetation and soil properties measured in the HRB in 2000 were used to replace the*
*universal BATS specifications, but the disturbances were not included in the simulation that*
*provided the data for this study. In the revision, we add discussion in section 4.2 about the*
*land-surface parameterization used in the model and the ways in which the disturbances*
*affect hydrological processes (Lines 477-486).*
7. All you describe in Section 3.2 is hydro-climatic characteristics and not the spatial
pattern of drought indices – as the section heading indicates and explained in text. See
the point 1 of my comment.
*Following the change described in the response to the comment 1, the term in heading is*
*changed from drought to aridity (Lines 150 and 282).*
8. The starting paragraph in the Introduction section does not flow - instead of making
a case (motivation) for the current study, it starts with describing the study catchment
(Heihe river) and then jumping to other region (Colorado river in US). Please reformulate.
*As suggested, the study catchment is presented later in the Introduction section, following the*
*descriptions of aridity indices and climate classification.*
9. The results section starts with detailing supplement plots – I would have expected
to see first the main plots and then the supporting plots and not other way around.
*These figures provide the climate background in the study region as well as simulation*
*validation, which we think is useful to understand the aridity results and validate model*
*performance. The concern with the comment may be partially due to the fact that we put*
*these figures in the supplement. In the revision, we remove the supplement and present all*
*figures in the main context. Some figures are combined to limit the number of figures.*
10. Figure 1: Quality is too poor – I could not read any of the subplot's legend.
*The first and second subplots of this figure are removed and the remaining subplot is more*
*readable.*
11. Figures 2-7: What is the point of making these figures up to 43oN, when the whole
study area ends at 42oN? Also please consider improving these figures and limit them
to just show the study basin region.
*As suggested, the upper portion of these figures is removed.*

**Major changes**

1. A paragraph is added to discuss the significance of this study for the research on the Heihe River Basin.
2. We use aridity indices to replace drought indices and clarification is provided.
3. SPI analysis and a brief description of model parameterization are provided.
4. Some writings of contexts or the way to present them are changed
5.  Figures are improved.
6. Author order is changed and a co-corresponding author is added. Both corresponding authors have made major contributions to research and manuscript writing and will contribute to meeting the production requirements if the paper is eventually accepted for publication.

[revised manuscript text omitted]

---

## Author Response (AR2)

**Responses to Reviewer #2**
(Responses in Italic)

Research on aridity is essential in these days where water and soil are natural resources to be considered in danger in several areas around the world. Aridity at local level is very interesting as the rapid changes in land cover and land use takes place and it is a constant in many areas.

In this manuscript, Liu et. al. analyzed meteorological and hydrological aridity indices to identify transitional climate zone in a Heihe River Basin (HRB) in the arid northwestern China. The authors used simulations from a Regional Integrated Environmental Model System (RIEMS 2.0). Based on their analyses, the authors found the hydrologic based aridity index being more suitable for the characterizing the transitional climate zone in the study basin, compared to meteorological based aridity index. I believe that provide valuable information in arid and subarid regions and can be connected to drought too.

*Thank the reviewer for reviewing our manuscript and providing very constructive and valuable comments and suggestions.*

Major issues:
The aridity index are used by the authors to climate characteristics (the seminal work of Budyko). Why the authors are using the full range of aridity index – which considers both wet and dry phase – in their analyses?. Could they focus more in different parts and then compare both indexes?

*Although the Heihe River Basin is located in the arid northwestern China, its upper reach actually has a humid climate because of the high elevations. The purpose of this study is to classify climate types in different Heihe River reaches, especially identifying a transition climate zone. This is why we analyzed the full range of the two indices, which cover both arid and humid climate types. The suggestion to focus more on different parts and then compare both indices is very valuable. Following this suggestion, we used new scale levels for the two aridity indices in Figs 6 and 7 so that each color represents a climate type. We also link the average aridity values with climate types in Table 3. The results in the figures and table are described and compared focused on different parts related to specific climate types.*

Authors are using at some point Standardized precipitation index (SPI) that is mainly for drought. Which is the relation between drought and aridity? Then, why don't you include the Standardized Precipitation Evapotranspiration Index (SPEI). Please specify if you go for droughts or for aridity, for both. By the way, SPI is not explained in Material and Methods.

*According to a critical review comment to our draft regarding the our confusion usage of the aridity and drought index terms, we tried to modify the draft and indicate it clearly in the previous revision that this study was to go with aridity indices. In the introduction section of this revision, we focused on aridity index description by (1) describe the aridity indices first, and (2) reducing the description of drought indices.*

*The analysis of SPI was added in the previous revision according to another review comment to our draft. SPI is a drought index. We kept it in the paper not because using it to analyze climate classification; instead, it was used to evaluate RIEMS performance in simulating meteorological drought. This is explained in this revision L253-254). The method to calculated SPI is described (L252-253).*

As far as I know, the RIEMS model provide estimates of net radiation (Rn) through their numerical parameterization of the mass, momentum and energy conservations schemes. Why not to use Rn variable and compare with the results when you use PET?

*The reviewer is right that Rn is estimated in RIEMS. The results were used in the calculation of PET with the Penman-Monteith method (Eq.1). We added analyses of Rn spatial pattern, seasonal cycle, and inter-annual variability in Figs. 2-4, and compared them with the corresponding results of other simulated variables.*

Why did you choose these indexes and not others that are available? Could you do a brief review on them.

There were two major considerations. First, the Budyko-type meteorological aridity index (AI) is a typical water-balance based index. It is simple but widely used in climate classification. ESI is a relatively new index. It is similar to AI but more related to surface hydrology. Second, these two indices reflect the water (precipitation and evapotranspiration) and heat (radiation) properties on the ground surface without the needs to obtain the complex vegetation and soil hydrological properties. The surface properties could be obtained from regional climate modeling, which was used in this study to obtain information for aridity index calculation. The reasons are stated in the revision (L117, L136-140). We also added some background about aridity indices in the first two paragraphs of the introduction section.

Finally, I am missing some statistics here, you put average, SD, CV but there is no test showing statistically significant differences among the three areas you describe in the HRB

*This is a very valuable suggestion. We added significant tests (L228). The results indicate that the averages of the aridity indices 
[revised manuscript text omitted]
 colors for AI and ESI indicate arid (grey), semi-arid (green), semi-humid (yellow) and humid (red) climate.

301302

302303     *PET* calculated using the Hamon method has the same pattern as the one using the Penman-

303304 Monteith method, but with smaller magnitude (Fig. 7). *PET* is mostly about 1 mm/d in the

304305 upper basin and increases to about 1.5-1.75 mm/d in the middle basin, and further to 1.75-2.25

305306 mm/d in the lower basin.

306307     The different spatial patterns between *AI* and *ESI* seen above are also found for the Homan

307308 method. *AI* is mostly above 0.6 in the upper basin (Fig. 7). It is below 0.2 in the middle and

308309 lower basins without apparent differences between the two regions. In contrast, while *ESI*

309310 remains large values of mostly above 0.9 in the upper basin and low values of below 0.2 in the

310311 lower basin, the values in many areas of the middle basin are 0.4-0.9, much different from

311312 those in the lower basin (Fig. 7).

312313

314316 Figure 7. Spatial distributions of potential evaporation (*PET*, mm/d), Aridity index (*AI*) and

315317 Evaporative Stress Index (*ESI*) with *PET* estimated using the Hamon method. Averaged  over

316318 1980-2010. The Heihe River basins are shown in the left panel. The colors for AI and ESI

indicate arid (grey), semi-arid (green), semi-humid (yellow) and humid (red) climate.

317319

**318320 3.3 Climate classification**

[revised manuscript text omitted]

---

## Author Response (AR3)

September 9, 2019

Dear Dr.  Schröter,

Thank you for your great efforts in evaluating our manuscript and kindly accepting it for publication. I am submitting the final version for production. Following the correction request, we added latitude and longitude lines in Fig.1.

Best regards,

Yongqiang Liu

USDA Forest Service

Yongqiang.liu@usda.gov